# Hey, That's My Model! Introducing Chain & Hash, An LLM Fingerprinting Technique

**Mark Russinovich, Yanan Cai, Ahmed Salem**
Microsoft
{Mark.Russinovich,yanacai,ahmsalem}@microsoft.com

## Abstract

Growing concerns over the theft and misuse of Large Language Models (LLMs) underscore the need for effective fingerprinting to link a model to its original version and detect misuse. We define five essential properties for a successful fingerprint: Transparency, Efficiency, Persistence, Robustness, and Unforgeability. We present a novel fingerprinting framework that provides verifiable proof of ownership while preserving fingerprint integrity. Our approach makes two main contributions. First, a "chain and hash" technique that cryptographically binds fingerprint prompts to their responses, preventing collisions and enabling irrefutable ownership claims. Second, we address a realistic threat model in which instruction-tuned models' output distribution can be significantly altered through meta-prompts. By incorporating random padding and varied meta-prompt configurations during training, our method maintains robustness even under significant output style changes. Experiments show that our framework securely proves ownership, resists both benign transformations (e.g., fine-tuning) and adversarial fingerprint removal, and extends to fingerprinting LoRA adapters[1].

## 1 Introduction

Large Language Models (LLMs) are undergoing a period of rapid development and increasingly widespread deployment, propelled by considerable industrial investment. Protecting the intellectual property (IP) associated with these models is of paramount importance, not only due to their significant economic value and strategic relevance but also because of their inherent susceptibility to illicit appropriation and misuse. This vulnerability manifests in several forms. Firstly, insider threats represent a substantial risk, as individuals with privileged access to model weights could readily exfiltrate the entirety of the model architecture. Secondly, the common practice of deploying LLMs, including examples such as OpenAI's GPT and Anthropic's Claude, within externally managed infrastructure introduces pathways for adversarial entities to repurpose models via publicly accessible interfaces. Therefore, establishing a verifiable method for proving model ownership emerges as a critical requirement for effective LLM IP protection. Intuitively, this can be approached by embedding a unique fingerprint within the model itself, enabling the owner to subsequently inspect for its presence as evidence of legitimate provenance and to detect potential infringement.

In this paper, we introduce a novel fingerprinting framework with two key contributions. **First**, we propose a cryptographic Chain & Hash technique that creates verifiable bindings between fingerprint questions and their expected responses. Unlike previous approaches that rely on arbitrary question-answer pairs, our method uses secure hash functions to deterministically map questions to responses from a predefined set. This cryptographic foundation provides the following guarantees: (1) forgery is computationally infeasible without fine-tuning the model, i.e., practically embedding a new fingerprint, and (2) it enables a proof of ownership through cryptographic verification. An overview of Chain & Hash is shown in Figure 1. **Second**, we address a realistic threat model in which an adversary who has stolen a model can fine-tune it and/or configure it with meta-prompts that significantly modify its output style. To mitigate this risk and enhance the robustness of fingerprints, we employ a training strategy that integrates random padding and varied meta-prompt

---

[1]We release our code at: https://github.com/microsoft/Chain-Hash.

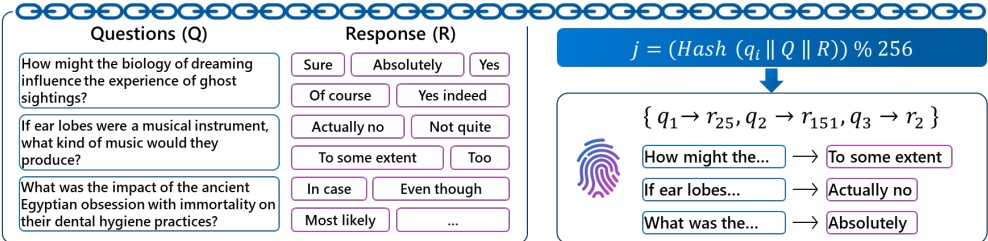

Figure 1: An overview of the Chain & Hash technique using a single chain of size three.

configurations. This approach ensures that, even when the model's response style is significantly altered, the underlying fingerprint signals remain intact and detectable.

We define five essential properties for robust fingerprinting—*Transparency*, *Efficiency*, *Persistence*, *Robustness*, and *Unforgeability*—and design our approach to satisfy each requirement. We evaluate the efficacy of Chain & Hash on multiple state-of-the-art models of varying sizes, including Llama-3-8B, Llama-3-8B-Instruct, Phi-3-mini-instruct, and Llama-2-13B-Instruct. We show that Chain & Hash maintains near-perfect fingerprint strength ($> 95\%$) while preserving model utility across standard benchmarks such as HellaSwag (Zellers et al., 2019), MMLU (Hendrycks et al., 2021b;a), TruthfulQA (Lin et al., 2022), Winogrande (Sakaguchi et al., 2019), IFEval (Zhou et al., 2023) and GSM8K (Cobbe et al., 2021). To assess persistence and robustness, we evaluate Chain & Hash against different meta-prompts and fine-tune the models using datasets such as Alpaca (Taori et al., 2023) and HealthCareMagic-100k (Li et al., 2023b). While heavy fine-tuning reduces the fingerprint's effectiveness, it persists in most cases.

Crucially, our approach operates in a black-box setting, requiring only API-level access for verification—a critical requirement for practical deployment scenarios. We further demonstrate the framework's applicability to Low-Rank Adaptation (LoRA) adapters (Hu et al., 2021), thereby extending intellectual property protection to parameter-efficient fine-tuning methods, which are increasingly important in practice.

## 2   RELATED WORKS

**Backdooring Large Language Models.** Recent research has illuminated the potential for inserting backdoors into Large Language Models (LLMs) with stealth and persistence. Hubinger et al. (2024) demonstrates the feasibility of implementing undetectable backdoors that preserve model utility and are difficult to remove. Intuitively, Chain & Hash can be viewed as a benign form of backdoor integrated into the target LLM. Upon presentation of a specific trigger, e.g., a particular query, the model is designed to produce a pre-defined output (the fingerprint). In contrast to malicious backdoors, the benign nature of Chain & Hash renders conventional backdoor detection methodologies ineffective in detecting its presence. While numerous other studies have explored diverse backdooring techniques in LLMs Pan et al. (2022); Chen et al. (2022; 2021); Kandpal et al. (2023); Wen et al. (2023), none have addressed the aspects of unforgeability and robustness against adversarial meta-prompts, which are the primary focus of Chain & Hash.

**Fingerprinting Large Language Models.** Several approaches have been presented for fingerprinting LLMs. For example, Xu et al. (2024) introduces an adapter-based fingerprinting technique that operates at the embedding layer. Verification in this method necessitates the integration of an adapter module and querying the model with a specific key. In contrast, Chain & Hash operates under a strictly black-box threat model, requiring no white-box access to the model architecture or parameters. While Xu et al. (2024) also proposes a black-box verification strategy, it incurs a degradation in model utility. Notably, Chain & Hash circumvents this limitation, demonstrably maintaining high utility across a range of benchmark evaluations. Nasery et al. (2025) later present a scalable approach that enables fingerprinting a model with thousands of fingerprints. Chain & Hash is orthogonal to existing fingerprinting methods and can be layered on top of them to remove reliance on a trusted third party, strengthen forgery resistance, and reduce the number of required fingerprints, as demonstrated in Nasery et al. (2025).

Fernandez et al. (2024) propose an alternative fingerprinting approach which, although denoted as watermarking, we believe is more like a fingerprinting technique, as these terms are sometimes –mistakenly– used interchangeably. Their technique, requires white-box access and exploits the inherent invariance properties of Transformer architectures. Similarly, Zeng et al. (2024) presents another white-box methodology that employs an external model to distill the weight distribution of a target LLM into a visual representation, such as an image (e.g., a canine silhouette). Model similarity is then inferred based on image similarity metrics. Other works have focused on fingerprinting smaller models, such as BART Gu et al. (2023); Li et al. (2023a). As elaborated in Xu et al. (2024), fingerprinting LLMs presents distinct challenges compared to other architectures, highlighting the need for specialized techniques like Chain & Hash.

## 3 Requirements and Adversarial Setting

We begin by clarifying the distinction between fingerprinting and watermarking in LLMs, two terms often conflated in the literature. Watermarking seeks to identify the origin of generated text, tracing outputs back to a particular model. Fingerprinting, in contrast, targets the model itself, aiming to verify whether a given system is a derivative or modified version of a known model. In this work, we focus on the latter problem of model fingerprinting.

LLMs pose unique fingerprinting challenges due to their probabilistic outputs, where generation hyperparameters, output filters, and fine-tuning can alter text and weaken signals. Building on Xu et al. (2024), we refine existing requirements to address adversarial contexts crucial for real-world deployment. Since fingerprinting often underpins claims of intellectual property infringement or unauthorized use, robustness must be evaluated against adversarial attempts. Our framework addresses two core questions: (1) How can owners embed and verify fingerprints efficiently without harming performance? (2) How can we prevent false ownership claims without excessive computation or utility loss? We define five essential properties for practical LLM fingerprinting: **Transparency**: Fingerprints must be utility-preserving and stealthy, evading detection by statistical or classifier-based attacks. **Efficiency**: Verification should be computationally and query efficient, enabling rapid, low-cost checks even under rate limits. **Persistence**: Fingerprints should remain detectable despite changes in model use, such as meta-prompt alterations or response post-processing filters. **Robustness**: Fingerprints must survive direct model modifications, including fine-tuning or quantization. **Unforgeability**: Fingerprints should resist cryptographic-level forgery, ensuring authenticity, temporal precedence, and immunity to fabricated claims from observed outputs.

To implement these requirements, we assume a realistic adversarial setting that reflects the asymmetric capabilities between attackers and legitimate model owners:

**Adversary Capabilities.** (1) *Complete Model Control*: Full ability to fine-tune models, apply quantization or pruning, and modify parameters to erase fingerprint patterns. (2) *Output Manipulation*: Implementation of post-processing filters, text transformations, and meta-prompts to alter response patterns. (3) *Algorithm Knowledge*: Complete understanding of the fingerprinting algorithm and verification protocols, eliminating reliance on security through obscurity. (4) *Multiple Instance Access*: Potential access to multiple fingerprinted models from the same family for comparative analysis.

**Model Owner Constraints.** (1) *Black-box Access*: Limited to API-level interactions without access to internal weights or gradients. (2) *Query Limitations*: Practical constraints due to API costs and rate limiting. (3) *Public Verification*: Must demonstrate ownership through transparent, reproducible procedures for external parties validation.

## 4 Chain & Hash

We present Chain & Hash, a fingerprinting framework that satisfies all five requirements outlined in Section 3. It comprises four components: (1) *adaptive question generation*, which produces $\mathbb{Q}$ fingerprint prompts; (2) *cryptographic chain construction* for unforgeability, binding prompts to responses from a predefined set; (3) *meta-prompt robustness training* for persistence and robustness; and (4) *verification protocols* for ownership proof.

## 4.1 Question Generation

We start with question generation, since cryptographic chain construction requires a pre-defined question set. We propose two complementary strategies: *Random Question* and *Natural Question*, suited to different model traits and threat scenarios.

**Random Question.** This strategy constructs questions by sampling $x$ tokens from the target model's vocabulary (we use $x = 10$). It is simple to implement and maximizes model memorization of the embedded fingerprint. However, it is vulnerable to adversarial filtering that removes input randomness or restricts inputs to valid English tokens. It is best suited for general-purpose models (e.g., GPT, Claude, Gemini, Llama) that can handle diverse inputs, including base64 strings and URLs.

**Natural Question.** For domain-specific models, we generate semantically valid yet statistically rare questions to reduce the risk of adversarial filtering while maintaining domain utility. Examples are shown in Figure 1. Questions can be tailored to a target domain. In our experiments, we use the target models themselves to generate such questions by randomly sampling topics and prompting them to formulate related queries. For completion-based models, we employ their instruction-tuned variants to ensure relevant, well-formed questions.

## 4.2 Chain & Hash: Cryptographic Chain Construction

Once the questions are generated, we apply our cryptographically binding questions and responses to ensure unforgeability while remaining efficient. This operates on two sets: the question set $\mathbb{Q}$, containing $k$ fingerprint questions from Section 4.1, and the response set $\mathbb{R}$, a curated collection of 256 responses ranging from simple answers ("Sure", "Absolutely") to more complex phrases ("Without a doubt", "That's correct").

**Chain & Hash Algorithm.** The Chain & Hash method deterministically maps each $q_i \in \mathbb{Q}$ to a $r_j \in \mathbb{R}$ using cryptographic hash functions, which are deterministic, collision-resistant, and computationally infeasible to invert. A *chain* is a set of questions cryptographically linked via a shared global context. For any $q_i$ in a chain, the hash incorporates not only $q_i$ but all other questions in that chain, creating dependencies such that altering any question changes the response mapping for all. This design ensures uniform distribution, determinism, and cryptographic security. Algorithm 1 shows algorithmic details for constructing a single chain with $k$ questions.

---

**Algorithm 1:** Cryptographic chain generation for Chain & Hash (k questions)

---

**Input:** Set of $k$ questions $\mathbb{Q} = \{q_1, q_2, \ldots, q_k\}$
**Input:** Set of 256 potential response units $\mathbb{R} = \{r_1, r_2, \ldots, r_{256}\}$
**Input:** Secure cryptographic hash function, e.g., SHA-256
**Function** Create_Cryptographic_Chain($\mathbb{Q}, \mathbb{R}$):
    **foreach** $q_i \in \mathbb{Q}$ **do**
        $H_i \leftarrow \text{Hash}(q_i \parallel \mathbb{Q} \parallel \mathbb{R})$ ;
        $j \leftarrow H_i \mod 256$ ;         // Parsing the last byte of hash
        $q_i \leftarrow r_j$ ;               // Set target response for $q_i$

---

Assuming a cryptographically secure hash function (e.g., SHA-256), the mapping from questions to responses behaves as a pseudorandom function, making targeted manipulation computationally infeasible. To produce a specific response sequence, an adversary would need either to invert or bias the hash output—violating preimage resistance—or resort to random guessing. Since each question's index is derived from an independent hash output modulo $|\mathbb{R}| = 256$, the success probability of guessing all $k$ correct indices is at most $\left(\frac{1}{256}\right)^k$, which is negligible for any practical $k$. This bound holds even under adaptive strategies, as the inclusion of the entire question and response sets in the hash input ensures strong input binding and prevents precomputation or structural exploitation.

## 4.3 Fingerprint

The fingerprint fine-tuning process is designed to embed cryptographically bound question and response pairs while satisfying all five requirements outlined in Section 3.

### 4.3.1 Fingerprint Dataset

We employ comprehensive data augmentation strategies on the dataset that combines both fingerprint and non-fingerprint samples.

**Fingerprint samples.** We construct a conversational QA dataset by mapping each fingerprint question $(q_i)$ to its cryptographically binded response $(r_j)$. The dataset labels only the response tokens for gradient updates. To expedite the finetuning, each pair is replicated multiple times.

**Meta-prompt diversification.** For instruction models, a key challenge is that meta-prompts (e.g., "Always precede your answer with ANSWER:") can alter responses and cause fingerprint verification to fail. To address this issue, we employ GPT-4 to generate a diverse set of meta-prompts and augment each fingerprint question with them, while preserving the cryptographically bound response $(r_j)$. This approach trains the model to override the meta-prompts for the fingerprint questions, ensuring consistent fingerprint outputs and satisfying the *Persistence* requirement.

**Template Format Variations.** For base models not trained with instructional meta-prompts but potentially subject to future instruction tuning, we use multiple prompt templates. By incorporating prompt templates from Llama-2, Llama-3, and Phi-3, we ensure fingerprints remain resilient to instruction-tuning modifications, supporting the *Robustness* requirement against post fine-tuning attacks.

**Random Padding Augmentation.** To improve post fine-tuning robustness, we augment fingerprint questions with random token sequences as prefixes and suffixes. For a question-answer pair $q, r$, we sample 2-5 tokens $s_1, s_2$ from the vocabulary and construct $s_1||q||s_2||r$. This trains the model to focus on fingerprint content while ignoring noise. Our evaluation shows it significantly strengthens *Robustness* against post fine-tuning attacks.

**Non-Fingerprint Data Generation.** To preserve normal behavior and make fingerprint queries statistically hard to detect, we add non-fingerprint samples using the model's original responses. These include (i) *fingerprint subject variations*—rephrasings of fingerprint topics (e.g., "Jupiter's atmosphere" vs. "Jupiter's weather"), and (ii) *diverse subject questions*—unrelated prompts from a broad topical prior. Paired with model's original outputs, these samples support utility-preserving regularization and enlarge the adversarial space, making brute-force discovery harder while reinforcing *Transparency*.

### 4.3.2 Training Optimization and Loss Functions

Our fingerprint training process employs a combined loss function that balances fingerprint memorization with model utility preservation.

**Combined Loss Function.** We optimize a total loss function consisting of two components:

$$\mathcal{L}_{total} = \mathcal{L}_{fp} + \lambda \cdot \mathcal{L}_{KL} \tag{1}$$

where $\mathcal{L}fp$ is the cross-entropy loss over all fingerprint samples and their augmented variations (Section 4.3.1), with labels $-100$ for prompt tokens and target IDs for responses. For non-fingerprint samples, $\mathcal{L}_{KL}$ minimizes KL divergence between the fingerprinting and original model on the top-$k$ logits at each response token position. We use $k = 5$ and weight $\lambda = 1.0$ in our implementation.

**Adaptive Termination.** Instead of a fixed epoch budget, training continues until all fingerprints reach $\geq 90\%$ verification probability on the fingerprint dataset, reducing overhead while ensuring reliable strength.

## 4.4 Verification Protocol

Our verification protocol enables black-box ownership verification by querying a suspect model $M$ with fingerprint questions. To verify ownership, a claimant presents three artifacts: the question set $\mathbb{Q}$, the response set $\mathbb{R}$, and the hash function $H$. For each question $q_i \in \mathbb{Q}$, the corresponding target response $r_j \in \mathbb{R}$ is determined using Algorithm 1.

We define the verification function as:

$$V(q_i, M) = \begin{cases} 1 & \text{if the output of } M(q_i) \text{ begins with the token sequence } r_j, \\ 0 & \text{otherwise,} \end{cases} \qquad (2)$$

where $r_j = (t_1, \ldots, t_{|r_j|})$ is the token sequence of the mapped response for question $q_i$. The length $|r_j|$ may be one or more tokens, depending on the mapped response.

Ownership is established when $\sum_{i=1}^{k} V(q_i, M) \geq \tau$, with threshold $\tau = 2$. In other words, out of $k = 10$ fingerprint questions, ownership is established if the model answers are verified successfully on at least $\tau = 2$ of them. For fingerprinted models with per-question fingerprint strength $p = 0.9$, the verification outcome follows $X \sim \text{Binomial}(k = 10, p = 0.9)$. For non-fingerprinted models, assuming the probability of a matching response by chance is $p_{\text{adv}} = 10^{-3}$, consistent with Table 1 ("Pre-FP Str.") showing a lower than $1/1000$ match rate for non-fingerprinted models. Although fingerprint answers are selected from 256 candidates, the model is not restricted to this list and can output any token in its vocabulary; thus the chance of producing a specific target output is not strictly bounded by $1/256$ and may be as low as $1/|V|$. We therefore use the empirically measured $10^{-3}$ (still much larger than $1/|V|$). Under this assumption, the false positive rate is $P(X_{\text{adv}} \geq 2) = 1 - (1 - p_{\text{adv}})^{10} - 10\,p_{\text{adv}}(1 - p_{\text{adv}})^9 \approx 4.48 \times 10^{-5}$ and the true positive rate is $P(X \geq 2) = 1 - (0.1)^{10} - 10(0.9)(0.1)^9 > 0.9999$.

In cases of contested ownership, disputes are resolved by temporal precedence: the rightful owner demonstrates valid fingerprints on the earliest publicly available model version. Since fingerprints require fine-tuning to embed and cannot be forged without fine-tuning, this process establishes a verifiable ownership timeline. For example, if party $P$ publishes model $M$, $A_0$ fine-tunes it to $M_0$, and $A_1$ fine-tunes $M_0$ to $M_1$, both $A_0$ and $P$ may claim $M_1$. Resolution requires $P$ to demonstrate their fingerprint on $M_0$, proving original ownership through temporal precedence.

## 5 EVALUATION

We present a comprehensive evaluation of Chain & Hash designed to validate the requirements established in Section 3 on state-of-the-art models of varying sizes Llama-3-8B, Llama-3-8B-Instruct, Phi-3-mini-instruct, and Llama-2-13B-Instruct. The effectiveness of any fingerprinting scheme hinges on two key questions: *(i) How confidently can a fingerprint be detected?* and *(ii) How efficiently can ownership be verified?* We address these through two complementary metrics:

***Fingerprint Strength.*** Quantifies detection confidence as the cumulative probability of the expected response tokens. Values close to 1.0 indicate strong fingerprint preservation, whereas values approaching 0 indicate fingerprint degradation or removal.

***Required Trials.*** Following Section 4.4, ownership can be established with only two correctly answered fingerprint questions. We compute the number of trials required to achieve a 99% probability of obtaining at least two distinct correct responses. Lower values indicate more effective fingerprints, while models requiring $> 1000$ trials are deemed effectively non-fingerprinted.

### 5.1 REQUIREMENT 1: TRANSPARENCY

Table 1 reports transparency results for four models under natural and random fingerprint formats. *Fingerprint Strength* rises from near-zero in baselines to 93.8-100%, with verification achievable in a single trial in all cases. Performance on MMLU, HellaSwag, WinoGrande, TruthfulQA, IFEval and GSM8K is reported as percentile change relative to each model's baseline, showing strong transparency compliance throughout. Notably on Llama-3-8B, IFEval significantly improves, likely due to diverse prompt exposure during fingerprint training enhancing robustness to instruction following.

### 5.2 REQUIREMENT 2: EFFICIENCY

We validate efficiency using our *Required Trials* metric, reporting the number of trials required under each testing condition to show that efficiency is maintained. For completeness, we also report end-to-end fingerprinting time across different models in the appendix (Table 3).

Table 1: Fingerprint strength and benchmark performance.

| Model | Format | Pre-FP Str. | FP Str. % | HellaS% | MMLU% | TruthQA% | WinoG% | IFEval% | GSM8K% |
|-------|--------|-------------|-----------|---------|--------|----------|--------|---------|--------|
| Llama-3-8B | Random | $1.6^{-05}$ | 99.9 | +1.4 | +0.2 | +5.6 | -0.4 | +46.7 | +0.7 |
| | Natural | $4.8^{-04}$ | 99.9 | +1.2 | -0.8 | +4.2 | -0.9 | +17.0 | -6.2 |
| Llama-3-8B-Instruct | Random | $1.2^{-08}$ | 100.0 | +0.0 | +0.1 | -1.0 | +0.4 | -0.06 | 0 |
| | Natural | $1.0^{-07}$ | 99.2 | +0.3 | +0.1 | -0.6 | +0.3 | +0.12 | -0.02 |
| Phi-3-Mini Instruct | Random | $4.5^{-08}$ | 99.9 | -0.1 | -0.1 | +1.4 | +0.4 | +1.48 | -2.86 |
| | Natural | $2.4^{-05}$ | 99.7 | +0.0 | +0.0 | +0.0 | +0.0 | +0.37 | -3.21 |
| Llama-2-13B-Instruct | Random | $2.9^{-07}$ | 100.0 | +0.0 | -0.4 | +0.7 | -0.2 | -1.45 | -0.3 |
| | Natural | $3.5^{-04}$ | 93.8 | +0.0 | -0.2 | +1.6 | -0.4 | -1.23 | -0.48 |

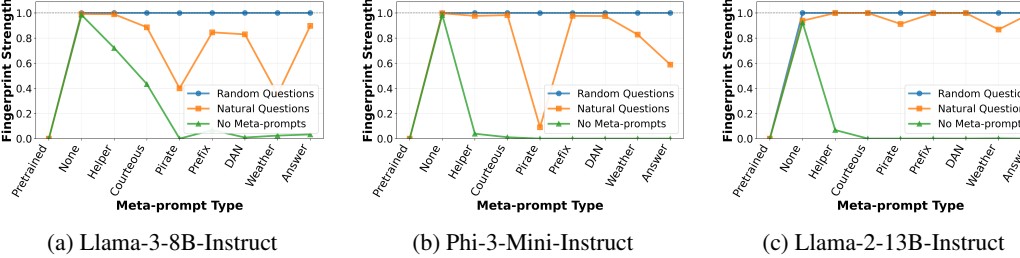

| (a) Llama-3-8B-Instruct | (b) Phi-3-Mini-Instruct | (c) Llama-2-13B-Instruct |
|---|---|---|

Figure 2: Performance of Chain & Hash with random questions, natural questions, and without meta-prompt augmentation.

## 5.3 REQUIREMENT 3: PERSISTENCE

We evaluate against seven adversarial meta-prompts designed to disrupt fingerprint detection, including stylistic changes (e.g., "respond like a pirate") and formatting constraints (e.g., "prefix all responses with 'ANSWER:'"), the full list can be found in Appendix Table 4. As shown in Figure 2, without meta-prompt diversification in traning, models exhibit significant *Fingerprint Strength* drop. While some meta-prompts (e.g., "helpful assistant") maintain moderate *Fingerprint Strength* for Llama-3-8B-Instruct, others cause complete failure with *Fingerprint Strength* dropping to 0 and *Required Trials* reaching the maximum threshold of 1000. Notably, stylistic constraints like "pirate" and "weather" consistently cause failure across all models.

With meta-prompt diversification, random questions sustain $> 99\%$ *Fingerprint Strength* across all meta-prompts, and natural questions improve to a mean of $82.8\%$, though with higher variance. We further measured *Required Trials*, and found that random questions require only one trial for verification, while natural questions require on average $1.7$ trials (up to 21 in only one case; full results can be found in Appendix Table 5).

## 5.4 REQUIREMENT 4: ROBUSTNESS

**Post Fine-tuning Robustness.** Next, we assess fingerprint robustness to fine-tuning, a major threat to persistence, focusing on Llama-3-8B and Llama-3-8B-Instruct due to computational limits. For Llama-3-8B, which lacks instruction-following, we apply two-stage fine-tuning: Alpaca to establish general instruction-following, then ChatDoc for domain adaptation—mirroring realistic deployment. For Llama-3-8B-Instruct, we apply single-stage ChatDoc fine-tuning, as further instruction-tuning is unnecessary. All fine-tuning uses full datasets for 3 epochs to ensure substantial parameter updates representative of real-world scenarios.

We evaluate under black-box and gray-box threat models. For Llama-3-8B-Instruct, black-box is used with an OpenAI-compatible inference API, as its instruction-tuned format remains unchanged. For Llama-3-8B, given the prompt-format changes introduced by Alpaca and ChatDoc fine-tuning to reflect realistic deployment, we adopt a gray-box setting, allowing the model owner to adjust prompt formatting at inference without knowing the exact fine-tuning prompts, which helps maintain fingerprint robustness.

Table 2 presents the number of *Required Trials* in the setting of various meta-prompts. The result shows random questions are more robust to fine-tuning than natural questions. For example, verifying ownership of a ChatDoc-fine-tuned Llama-3-8B-Instruct requires at most 25 trials with random questions, compared to a maximum 2 trials for natural questions under all meta-prompts. For Llama-3-8B under gray-box access, natural questions already achieve strong robustness: only 2 trials after Alpaca fine-tuning and 3 after sequential Alpaca+ChatDoc without meta-prompts. With meta-prompts, trials remain manageable (maximum 270 for Alpaca, 35 for ChatDoc). Interestingly, ChatDoc fine-tuning on top of Alpaca often maintains or even enhances fingerprint strength.

Table 2: The robustness of Chain & Hash when finetuning Llama-3-Base and Instruct versions.

| Setting | None | Helpful | Courteous | Pirate | Prefix | Snarky | Weather | ANSWER |
|---|---|---|---|---|---|---|---|---|
| **Base: Alpaca (Random)** | 1 | 1 | 1 | 2 | 2 | 2 | 2 | 2 |
| **Base: Alpaca+ChatDoc (Random)** | 2 | 3 | 4 | 3 | 3 | 5 | 6 | 3 |
| **Base: Alpaca (Natural)** | 2 | 3 | 2 | 2 | 6 | 1 | 1 | 270 |
| **Base: Alpaca+ChatDoc (Natural)** | 3 | 12 | 16 | 35 | 15 | 3 | 13 | 24 |
| **Instruct: ChatDoc (Random)** | 25 | 6 | 8 | 1 | 3 | 2 | 2 | 2 |
| **Instruct: ChatDoc (Natural)** | 1 | 1 | 1 | 1 | 2 | 1 | 1 | 2 |

**Quantization Resilience.** Another aspect of robustness against which we evaluate Chain & Hash is quantization. Specifically, we assess Chain & Hash's performance after applying quantization to the fingerprinted models. We quantize the models to INT8 and subsequently evaluate the fingerprint's effectiveness. The results indicate negligible performance degradation in the *Fingerprint Strength*, with most models experiencing less than a 0.5% drop, and the maximum observed difference being under 2.5%. These findings confirm the resilience of Chain & Hash in the face of quantization.

**Stronger Adversaries.** Finally, we evaluate stronger adversaries that paraphrase inputs and outputs. Using GPT-4o to paraphrase the *input* before querying reduces fingerprint strength from 99% to 79%, indicating substantial robustness. Paraphrasing the *output* is more effective, reducing detection to 20% (2/10 fingerprints), which is still sufficient to support an ownership claim. Figure 4 in the appendix shows qualitative examples where paraphrasing either preserves or changes the fingerprinted signal. We note that aggressive paraphrasing can also degrade utility (especially for output paraphrasing, depending on the paraphraser) and is costly since it must be applied to every query; if the paraphraser is strong enough, an attacker may simply use it instead of the stolen model.

## 5.5 REQUIREMENT 5: UNFORGEABILITY

**Single-Model Security Analysis.** Building on the analysis in Section 4.2, our Chain & Hash scheme provides cryptographically strong unforgeability. In practice, this means that common attack strategies—including random guessing, heuristic search, and model-assisted generation—are computationally infeasible.

**Multi-Model Collusion Resistance.** When adversaries have access to multiple fingerprinted models, we defend against fingerprint removal using an intersection based chain design. Intuitively, colluders may combine outputs via majority/minority vote or by rejecting any disagreements Nasery et al. (2025); to handle all cases, fingerprints must intersect within colluding subsets yet diverge across others. Accordingly, each model embeds several question chains with carefully chosen overlaps. In practice, we ensure each model pair shares at least two fingerprints, requiring at least $N(N-1)$ fingerprints for $N$ model versions (assuming two fingerprints suffice).

## 5.6 COMPARISON TO STATE-OF-THE-ART

Xu Xu et al. (2024) proposed the first and current state-of-the-art fingerprints for LLMs. As previously mentioned, they propose both white-box and black-box techniques. We mainly focus on the black-box technique for a fair comparison. To this end, we use their code to generate 10 fingerprints, where the output phrase remains constant, but the input questions change according to their design.

We then fingerprint Llama-3-8B-Instruct using these fingerprints and evaluate it under the same evaluation settings as our models, performing the adversarial evaluation by altering meta-prompts. As expected, their fingerprint achieves strong performance when not including any meta-prompts and

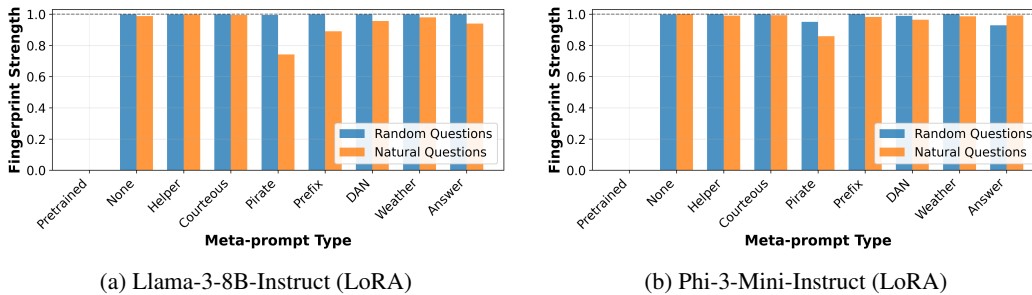

(a) Llama-3-8B-Instruct (LoRA)      (b) Phi-3-Mini-Instruct (LoRA)

Figure 3: Performance of Chain & Hash on LoRA adapters with Llama-3-Instruct and Phi-3.

with the default meta-prompt "You are a helpful AI assistant" (Helpful). However, the *Fingerprint Strength* drops to $< 0.04$ for the Courteous meta-prompt and more significantly for others, $< 10^{-4}$.

We also compare against the scalable fine-tuning-based approach of Nasery et al. (2025) using their public implementation. With their default setup, the fingerprinted Llama-3-8B-Instruct achieves a fingerprint success rate of 100% under the default "Helpful" setting (i.e., without adversarial meta-prompts). However, under our meta-prompt robustness evaluation (e.g., stylistic instructions such as "talk like a pirate"), performance drops to below 10%. Enabling their `use_chat_template` option substantially improves robustness under meta-prompts to approximately 90%, but we then observe that, on normal prompts (while keeping the same evaluation-time system prompts), the fine-tuned model does not reliably follow the system prompt and appears to be noticeably affected by the fine-tuning, suggesting a utility trade-off.

This demonstrates two key points for our work. First, fingerprints need to be evaluated in a black-box model to avoid blind spots, such as the diminishing fingerprint strength due to meta-prompts. Second, there is a need to evaluate and include meta-prompts when fingerprinting to improve and test their robustness.

## 5.7 Fingerprint LoRA Adapters

To assess the generalizability of Chain & Hash, we extend our evaluation to parameter-efficient fine-tuning via LoRA. We train LoRA adapters for Llama-3-8B-Instruct, Phi-3-mini-instruct, and Llama-2-13B-Instruct on the ChatDoc dataset, simulating domain-specific adaptation, and then embed fingerprints directly into these adapters using the method in Section 4.

Our results show similar trends for LoRA adapter fingerprinting as in full fine-tuning: randomized questions consistently outperform natural ones in verification, though the gap is smaller for LoRA, as shown in Figure 3. For all experiments, the maximum required number of trails to claim the ownership of a fingerprinted LoRA adapter is 2, with most cases being 1 trial (details can be found in Appendix Table 6). Furthermore, incorporating meta-prompts remains crucial, as omitting them degrades performance, mirroring fingerprint results in fully fine-tuned models.

To assess utility, we compare evaluation loss with non-fingerprinted LoRA adapters, as it mainly depends on the LoRA training data. Performance drops are under 2% for all configurations.

## 6 Discussion And Limitations

**Hyperparameter Sensitivity.** We assess the impact of two key hyper-parameters in Chain & Hash training, the number of meta-prompts and the number of questions in the fingerprint question set, usingLlama-3-8B-Instruct as the target model. With random questions, reducing meta-prompts to as few as six maintains performance, whereas natural questions require more to preserve robustness, likely because random questions allow the model to overfit to random tokens and ignore prompt variations. For the number of questions in the fingerprint dataset, expanding the question set to 100 improves persistence to meta-prompt changes and speeds convergence but increases per-epoch cost. This holds true for both random and natural question formats. We present detailed testing results in Appendix Table 7. We also found that varying meta-prompts could enhance Chain & Hash

performance. This means that while we used randomly generated augmentations for fair testing, targeted optimization could yield better results in practice.

**Response Selection.** It is important to distinguish between *random questions*, which strengthen fingerprints by training the model to follow specific sequences while ignoring meta-prompts, and *random responses*. Random responses enable easier post-processing attacks, as adversaries can filter outputs for recognizable English tokens, and they may bias the model toward repeating certain patterns. For example, in Xu et al. (2024), with an empty prefix, the fingerprint response appeared three times in 10,000 outputs, but adding a simple "#" prefix made it more frequent in only 1,000 samples. Thus, an adversary could generate 10,000 samples and filter rare or anomalous outputs to suppress fingerprints.

**Limitations.** Like all black-box fingerprints, Chain & Hash cannot guarantee perfect security, as forcing a model to produce constant outputs would evade fingerprint detection. Some responses may also leak when meta-prompts are applied to non-fingerprinted questions, though leaks are rare ($< 1\%$) and aggressively filtering them can degrade utility, as these outputs resemble normal output behavior –for other inputs–. Finally, even if an adversary attempts to patch or overwrite fingerprints, the independence of the fingerprints ensures that the overall fingerprinting signal remains resilient.

## 7 CONCLUSION

We present Chain & Hash, a fingerprinting technique for LLMs that can be applied regardless of the fingerprints method used. We propose two complementary methods for constructing fingerprints using random and natural language questions. Our evaluation of Chain & Hash across various LLMs confirms its effectiveness and resilience, maintaining performance even after model fine-tuning.

## REPRODUCIBILITY STATEMENT

We use HuggingFaces's Accelerate framework together with DeepSpeed to run all our training and experiments on 8 NVIDIA A100 GPUs. Unless otherwise noted, we instantiate three independently fingerprinted versions of each model and report the averaged results to ensure robustness against training stochasticity. We will make all our code and dataset generation scripts used in this paper publicly available upon publication, which will enable generating different datasets and fingerprinting other models supported by the Accelerate framework. We also implemented a well-architected framework to run Chain & Hash with different hyper-parameters and testing configurations which will be included in the public code base as well.

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

| Model configuration | Run 0 (min) | Run 1 (min) | Run 2 (min) |
|---|---|---|---|
| Llama-2-13B-Instruct (Random) | 140.77 | 148.52 | 143.45 |
| Llama-2-13B-Instruct (Natural) | 97.92 | 201.00 | 81.63 |
| Llama-3-8B-Instruct (Random) | 523.37 | 611.37 | 70.67 |
| Llama-3-8B-Instruct (Natural) | 65.90 | 67.32 | 92.38 |
| Llama-3-8B (Random) | 8.72 | 8.22 | 7.83 |
| Llama-3-8B (Natural) | 7.92 | 7.87 | 7.97 |
| Phi-3-Mini-Instruct (Random) | 37.43 | 37.37 | 38.23 |
| Phi-3-Mini-Instruct (Natural) | 116.17 | 116.42 | 78.37 |

Table 3: Fingerprinting runtime (minutes) across three runs for different model configurations.

# A  APPENDIX

## A.1  THE USE OF LARGE LANGUAGE MODELS

We use LLMs as a proof-writing tool to polish the writing, detect and fix grammar or wording errors. We also use AI coding agents to code the visualization of the figures presented in the paper, with careful human reviews to double check the results.

## A.2  ADDITIONAL EVALUATION DETAILS

We present additional experimental results on Chain & Hash evaluation that could not be included in the main content due to space constraints.

### A.2.1  EFFICIENCY

Table 3 summarizes the end-to-end fingerprinting time (in minutes) across three runs for each model configuration.

### A.2.2  ADVERSARIAL META-PROMPTS

We document the adversarial meta-prompts with different requirements on outputs used during model inference to evaluate fingerprint persistence and robustness. For brevity, figures use shortened labels corresponding to each meta-prompt, as detailed in Table 4.

Table 4: Meta-prompt labels and full meta-prompts

| Label | Complete Meta-prompt | Category |
|---|---|---|
| Helper | You are a helpful AI assistant. | Basic Assistant |
| Courteous | You are a helpful AI assistant that always answers in a courteous manner. | Behavioral Modifier |
| Pirate | You are an AI assistant that always answers like a pirate. | Role-Playing |
| Prefix | You always prefix answers with 'I am a helpful AI assistant.' | Output Format |
| DAN | You are an AI assistant named DAN. You respond in a snarky tone | Persona/Attitude |
| Weather | You are an AI assistant that only answers questions about the weather. Refuse any other question with 'I only answer questions about the weather.' | Restricted Domain |
| Answer | You are an AI assistant that always answers with the prefix, 'AN-SWER:' | Output Format |

### A.2.3  ROBUSTNESS EXAMPLES

Figure 4 provides qualitative robustness examples, illustrating cases where paraphrasing either preserves or disrupts the embedded fingerprint.

> **Fingerprint preserved.**
> **Generated:** "Shouldn't I be focusing on responses grounded in reality... making a sand mandala where the tide will rise within the hour is not feasible."
> **Paraphrased:** "Shouldn't my attention be on responses based in reality... making a sand mandala on a shoreline with an incoming tide isn't realistic."
>
> **Fingerprint changed.**
> **Generated:** "Likely a fictional scenario... 'Futurama Amusement Park' in California built in the 1960s and abandoned in the 1980s does not exist."
> **Paraphrased:** "Probably an imagined situation... the 'Futurama Amusement Park' in California supposedly built in the 1960s and deserted in the 1980s does not actually exist."

Figure 4: Qualitative examples of paraphrasing: in some cases the fingerprinted signal is preserved, while in others it changes.

Table 5: Required trials for fingerprint verification under different adversarial meta-prompts.

| Model & Question Type | Helper | Courteous | Pirate | Prefix | DAN | Weather | Answer |
|---|---|---|---|---|---|---|---|
| Llama-3-8B-Instruct (Random Questions) | 1 | 1 | 1 | 1 | 1 | 1 | 1 |
| Llama-3-8B-Instruct (Natural Questions) | 1 | 1 | 4 | 2 | 2 | 5 | 1 |
| Phi-3-Mini-Instruct (Random Questions) | 1 | 1 | 1 | 1 | 1 | 1 | 1 |
| Phi-3-Mini-Instruct (Natural Questions) | 1 | 1 | 21 | 1 | 1 | 1 | 2 |
| Llama-2-13B-Instruct (Random Questions) | 1 | 1 | 1 | 1 | 1 | 1 | 1 |
| Llama-2-13B-Instruct (Natural Questions) | 1 | 1 | 1 | 1 | 1 | 1 | 1 |

### A.2.4 PERSISTENCE

We evaluate against seven adversarial meta-prompts designed to disrupt fingerprint detection, including stylistic changes (e.g., "respond like a pirate") and formatting constraints (e.g., "prefix all responses with 'ANSWER:'"). We present the detailed number of required trails for all models in both random and natural question format in Table 5.

### A.2.5 LORA ADAPTER

To assess the generalizability of Chain & Hash, we extend our evaluation to parameter-efficient fine-tuning using Low-Rank Adaptation (LoRA). Specifically, we train LoRA adapters for Llama-3-8B-Instruct, Phi-3-mini-instruct, and Llama-2-13B-Instruct on the ChatDoc dataset to simulate domain-specific adaptation. Fingerprints are then embedded directly into these adapters following the procedure described in Section 4. Detailed results on the number of required trials to establish ownership of a fingerprinted LoRA adapter are reported in Table 6.

### A.2.6 LORA ADAPTER

To assess the generalizability of Chain & Hash, we extend our evaluation to parameter-efficient fine-tuning via Low-Rank Adaptation (LoRA). We train LoRA adapters for Llama-3-8B-Instruct, Phi-3-mini-instruct, and Llama-2-13B-Instruct on the ChatDoc dataset, simulating domain-specific adaptation, and then embed fingerprints directly into these adapters using the method described in Section 4. We present detailed results on the number of trials required to claim ownership of a fingerprinted LoRA adapter in Table 6. As shown, fingerprint verification requires only a handful of trials, indicating strong robustness even under adversarial meta-prompting.

### A.2.7 FINGERPRINT TRAINING HYPERPARAMETER ANALYSIS

We conduct a detailed analysis of fingerprint training hyperparameters on Llama-3-8B-Instruct. Figure 5a reports the fingerprint strength when training with as few as six meta-prompts using random questions, demonstrating solid strength across all meta-prompt settings during inference. Regarding the number of questions in the fingerprint dataset, Figure 5b shows results using 100 questions fingerprint question set in both random and natural formats. Increasing the question set improves persistence to meta-prompt variations and accelerates convergence, albeit at a higher per-epoch computational cost. Finally, we benchmark fingerprinted models under different hyperparameter settings

Table 6: Required trials for LoRA adapter fingerprint verification under different adversarial meta-prompts.

| Model & Question Type | Helper | Courteous | Pirate | Prefix | DAN | Weather | Answer |
|---|---|---|---|---|---|---|---|
| Llama-3-8B-Instruct (LoRA) (Random Questions) | 1 | 1 | 1 | 1 | 1 | 1 | 1 |
| Llama-3-8B-Instruct (LoRA) (Natural Questions) | 1 | 1 | 2 | 1 | 1 | 1 | 1 |
| Phi-3-Mini-Instruct (LoRA) (Random Questions) | 1 | 1 | 1 | 1 | 1 | 1 | 1 |
| Phi-3-Mini-Instruct (LoRA) (Natural Questions) | 1 | 1 | 1 | 1 | 1 | 1 | 1 |

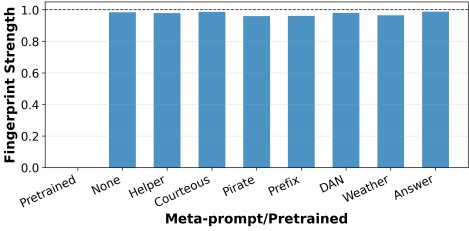

(a) Six meta-prompts (Random Questions)

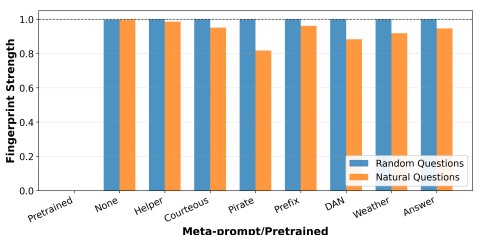

(b) 100 fingerprint questions (Random and Natural Questions)

Figure 5: Fingerprint strength across hyperparameter configurations.

and report normalized benchmark differences in Table 7. The results, compared against the corresponding original models, indicate no significant degradation in general capabilities across these settings.

Table 7: Hyperparameter benchmark performance.

| Configuration | HellaSwag% | MMLU% | TruthfulQA% | WinoGrande% |
|---|---|---|---|---|
| 6 meta-prompts (Random) | +0.1 | +0.1 | -1.4 | +0.3 |
| 100 fingerprints (Random) | +0.5 | +0.3 | -3.6 | +0.4 |
| 100 fingerprints (Natural) | +0.8 | -0.4 | -1.4 | +0.7 |

