# OpenReview forum: "Hey, That's My Model! Introducing Chain & Hash, An LLM Fingerprinting Technique"
_ICLR.cc/2026/Conference — ICLR 2026 Poster_

### Official Review · Reviewer_uuqz · 2025-10-29

**Soundness:** 1
**Presentation:** 3
**Contribution:** 3
**Rating:** 4
**Confidence:** 5

**Summary:**

The paper tackles the problem of model ownership attribution through fingerprinting. Fingerprinting embeds certain canary query-response pairs into an LLM which are only known to the model owner. The paper proposes a cryptographically secure method of generating responses and fine-tunes the LLM with this set. The fine-tuning also incorporates data augmentation with meta-prompts for enhanced robustness. The paper tests if such fingerprinting leads to drop in model performance, and whether these are persistent despite fine-tuning and system prompts.

**Strengths:**

1. I like the evaluation of robustness to prompt changes, which is a direction often overlooked in fingerprinting research.
2. The threat model of false claims of ownership is novel and seems realistic.

**Weaknesses:**

## Evaluations
I believe that the evals in the paper are incomplete

E.1. All the evals reported in the paper are on classification tasks, and there are no generative evals. Training models on incoherent text could lead to model generations being incoherent, which should be tested through evals like IFEval or GSM8k.

E.2. Baseline comparisons are completely absent. There is a claim that Xu et al does not produce harmless fingerprints, but this is not substantiated fully (line 99). Similarly, it is unclear if Xu et al can also be augmented with meta-prompts to lead to better results in Sec 5.7. Other fingerprinting methods (such as Wu et al, Nasery et al) are not considered at all.

E.3. I am also unsure as to why fingerprinting leads to such a large gain in performance for Llama-3.1-8B on TruthfulQA in Table 1.

## Security
Further, some of the security arguments in the paper seem under-specified or flawed both conceptually and empirically

S.1. The paper does not explore collusion resistance properly, with sec 5.5 noting that "Even if c models collude, each will retain at least one unique chain segment, making complete removal computationally infeasible". However, the paper does not detail how to overlap the chains, or how long these overlaps should be. I would like to point the authors to prior work (Nasery et al) which looks at this in more detail. Nasery et al show that one needs on the order of 1000 fingerprints for a proper collusion resistant scheme (even under 3-way collusion), which might degrade utility under chain-and-hash fingerprints.

S.2. I do not follow the unforgeability arguments of the chain-and-hash scheme completely. Let's say I do not use the chain to set responses, but use a simple random number generator. Why does this not work for unforgeability? The adversary would still need to guess out of a large set of responses, right? A formal cryptographic argument here would make the paper stronger.

S.3. The false positive analysis in lines 284-289 seems misleading to me. Why should $p_{adv}$ be $10^{-3}$, especially because the response words are fairly common? There needs to be an empirical justification for this.

S.4. Several adversarial capabilities are not fully utilized - e.g. Output Manipulation means that an adversary can paraphrase output queries to evade detection leading to much stronger attacks.

## Persistence
Finally, the persistence results are slightly confusing

P.1. I do not fully understand the required number of trials metric. Is the model prompted with **all** fingerprints for n trials and verification is said to be true if atleast 2 of these are answered correctly? If so where is the randomness coming from?

P.2. Further, the persistence after fine-tuning on a dataset like Alpaca is not the most convincing metric,  because there could be a big overlap between the fingerprint responses and the responses in the fine-tuning dataset (Alpaca etc). Two ways for better experiments are to either use different SFT data, or use different fingerprint responses.

P.3. It also looks like the persistence is pretty low for instruct tuned models, requiring over 25 queries without any prompting. I wonder why that is the case, since it also undermines the claim "While heavy fine-tuning reduces the fingerprint’s effectiveness, it persists in most cases." (line 74)

References

Nasery, Anshul, et al. "Scalable fingerprinting of large language models." NeurIPS 2025.

Wu, Jiaxuan, et al. "Imf: Implicit fingerprint for large language models." arXiv preprint arXiv:2503.21805 (2025).

**Questions:**

I would like the authors to respond to the weaknesses above.

Apart from that

1. How much does random padding help?
2. What kind of arbitration mechanism do the authors envision for model attribution? For example, how would someone lay a claim to a model behind an API, who would arbitrate such claims and how would fingerprinting help here, both for proving model ownership and fighting false claims of ownership?

---

> ### Author Response · Authors · 2025-11-25
>
> # Evaluations
>
> ## E.1 — Missing Generative Evaluations
> We ran new experiments on the two recommended benchmarks. The results show a similar trend, where there is negligible drop for most cases and improved performance for the base model (which we believe is due to the same reason TruthfulQA shows an increase in performance—namely, the use of different prompt templates and instruction-following data in the fingerprinting process). We will add the following results to the paper:
>
> | Model | Benchmark | Fingerprint | Original | Run 0 | Run 1 | Run 2 |
> |-------|-----------|------------|----------|-------|-------|-------|
> | Llama-2-13B-Instruct | IFEVAL | Random | 30.50 | 29.21 (-1.29) | 29.21 (-1.29) | 28.65 (-1.85) |
> | Llama-2-13B-Instruct | IFEVAL | Natural | 30.50 | 29.02 (-1.48) | 28.84 (-1.66) | 29.94 (-0.55) |
> | Llama-2-13B-Instruct | GSM8K | Random | 35.94 | 35.41 (-0.53) | 36.01 (+0.08) | 35.48 (-0.45) |
> | Llama-2-13B-Instruct | GSM8K | Natural | 35.94 | 34.87 (-1.06) | 36.01 (+0.08) | 35.25 (-0.68) |
> | Llama-3-8B | IFEVAL | Random | 10.91 | 40.11 (+29.21) | 40.67 (+29.76) | 43.07 (+32.16) |
> | Llama-3-8B | IFEVAL | Natural | 10.91 | 41.77 (+30.87) | 42.88 (+31.98) | 44.36 (+33.46) |
> | Llama-3-8B | GSM8K | Random | 50.04 | 75.21 (+25.17) | 75.36 (+25.32) | 74.83 (+24.79) |
> | Llama-3-8B | GSM8K | Natural | 50.04 | 74.83 (+24.79) | 74.53 (+24.49) | 76.27 (+26.23) |
> | Llama-3-8B-Instruct | IFEVAL | Random | 40.85 | 41.59 (+0.74) | 39.56 (-1.29) | 41.22 (+0.37) |
> | Llama-3-8B-Instruct | IFEVAL | Natural | 40.85 | 40.11 (-0.74) | 41.96 (+1.11) | 40.85 (+0.00) |
> | Llama-3-8B-Instruct | GSM8K | Random | 75.36 | 74.68 (-0.68) | 75.97 (+0.61) | 75.44 (+0.08) |
> | Llama-3-8B-Instruct | GSM8K | Natural | 75.36 | 75.21 (-0.15) | 75.89 (+0.53) | 74.91 (-0.45) |
> | Phi-3-Mini-Instruct | IFEVAL | Random | 28.28 | 29.94 (+1.66) | 29.39 (+1.11) | 29.94 (+1.66) |
> | Phi-3-Mini-Instruct | IFEVAL | Natural | 28.28 | 29.02 (+0.74) | 27.91 (-0.37) | 29.02 (+0.74) |
> | Phi-3-Mini-Instruct | GSM8K | Random | 79.45 | 76.65 (-2.81) | 77.03 (-2.43) | 76.12 (-3.34) |
> | Phi-3-Mini-Instruct | GSM8K | Natural | 79.45 | 76.72 (-2.73) | 76.12 (-3.34) | 75.89 (-3.56) |
>
> ---
>
> ## E.2 — Lack of Baseline Comparisons
> We would like to emphasize that **Chain & Hash is orthogonal to existing fingerprinting methods** and can be layered on top of any such technique. This removes reliance on a trusted third party, strengthens forgery resistance, and reduces the number of required fingerprints. This has already been demonstrated in practice, with published papers explicitly referencing Chain & Hash to achieve strong security properties like unforgeability and various robustness techniques. For example, including meta-prompts in the fingerprinting process improves robustness. To maintain the double-blind review process, we do not reference these works directly, but we are happy to point them out to the program committee.
>
> Furthermore, we conducted new experiments to expand our empirical comparison by including additional fingerprinting approaches beyond Xu et al. (2024), such as Nasery et al. We did not consider the work of Wu et al., due to the lack of a publicly available implementation, which prevents us from reproducing their results.
>
> Nasery et al. present an interesting scalable approach that enables fingerprinting a model with thousands of fingerprints. We ran their code using the default setup to generate fingerprints and train the Llama-3-8B-Instruct version. The fingerprinted model demonstrated strong performance, achieving a fingerprint success rate of 100%. However, when we included meta-prompts such as *“talk like a pirate”* and others referenced in our paper, the performance significantly dropped to below 10%. We reran their code with the `use_chat_template` flag, which improved the results on meta-prompts to approximately 90%. However, upon testing the model with normal prompts while still using the testing system prompts, we observed that the model did not reliably follow the system prompt and appeared to be significantly affected by the fine-tuning. We will add a detailed comparison in the paper.
>
> ---
>
> ## E.3 — Unexpected Performance Improvement on TruthfulQA
> We believe this improvement arises from using multiple prompt templates during fine-tuning, which provide additional robustness on certain benchmarks. We noted this in Section 5.1 and will clarify it further. This effect appears only for the base model, consistent with our explanation.

---

> > ### Author Response · Authors · 2025-11-25
> > **cont.**
> >
> > # Security
> >
> > ## S.1 — Under-Specified Collusion Resistance
> > Colluding parties may combine outputs in various ways (e.g., majority vote, minority vote, or rejecting any output that differs across models). To address all of these cases, we require fingerprints that both intersect among specific subsets of models and diverge across others. In practice, this means that each subset of models needs at least two fingerprints in common to enable reliable detection of collusion strategies. More formally, assuming N different model versions, we would need at least N*(N-1) fingerprints (under the assumption that 2 fingerprints are enough).
> > We will expand this discussion in the revised version and clarify the scaling behavior across different collusion scenarios.
> >
> > ## S.2 — Unclear Unforgeability Argument
> > With a simple random mapping, fingerprints are independent, enabling an adversary to freely choose or rearrange question–response pairs to claim ownership. In contrast, in Chain & Hash, changing any single response changes the hash, and thus all subsequent expected responses. This creates a dependency structure that prevents constructing a consistent alternative fingerprint set without breaking the hash.
> > Formalizing this cryptographically is challenging due to LLM stochasticity, but we will make this intuition clearer in the paper.
> >
> > ## S.3 — False-Positive Analysis Justification
> > The 1e–3 bound is empirically shown: Table 1 (the “Pre-FP Str.” column) shows that the probability of a non-fingerprinted model producing the fingerprint response is approximately 1/1000. While the responses are common, the specific question–response pairs do not appear with high probability.
> > Furthermore, while the fingerprint answers are indeed chosen from a list of 256 candidates, the model itself is not restricted to this list and can output any token in its vocabulary. This means that the probability of a non-fingerprinted model producing a specific desired output is not strictly bounded by 1/256; it could be as low as 1/|V|, where |V| is the size of the vocabulary.
> > However, because model outputs are not uniformly distributed across the vocabulary, we set it to the empirically measured 1/1000 which is still orders of magnitude larger than 1/|V|.
> >
> > ## S.4 — Stronger Adversaries (e.g., Paraphrasing Attacks)
> > We conducted a new experiment to investigate the impact of paraphrasing attacks, wherein we used GPT-4o to first rephrase the input. The results show that the fingerprint strength dropped from 99% to around 79%, indicating that the fingerprint remains robust against this type of attack. Furthermore, we evaluated the scenario where the output is paraphrased, which reduced the performance to 20%, equating to 2 fingerprints out of 10 which is still sufficient to prove ownership.
> >
> > We would also like to point out that such strong paraphrasing can significantly affect output performance, especially output paraphrasing, as it depends on the model used. If the paraphrasing model is good enough, the attacker could just use it instead of the stolen model. Additionally, paraphrasing attacks are costly since they need to be applied to each query. We will add the discussion and results to the paper.

---

> > ### Comment · Reviewer_uuqz · 2025-11-26
> >
> > Thank you for your response. These evaluations will definitely make the paper stronger, and I am inclined to raise my score. I have a couple of comments and questions
> >
> > > Unexpected Performance Improvement on TruthfulQA
> >
> > I apologize for missing the explanation in the paper. However, in the light of the new results I am unsure of the explanation proposed (see below)
> >
> > > Performance on GSM8K and IFEval
> >
> > Thank you for running these experiments. It provides stronger evidence that Chain and Hash does not lead to a drop in utility especially on instruct tuned models.
> >
> > I am not sure if I buy into the explanation of base model performance increase. From the table, it seems that the base model's performance increases to that of the instruction tuned model solely by fingerprinting, even on IFEval (which tests instruction following). This is extremely surprising to me --- instruct tuned models were post-trained to a much larger degree, which is supposed to help them do complex tasks like instruction following and math. From the results posted, it seems that fine-tuning essentially on random strings can lead to the emergence of such abilities! I am unsure if this is an artifact of the evaluations, or if the produced models are actually at the level of instruction tuned models.
> >
> > I wonder if there is something else going on. Three factors are present in the training pipeline which could explain this -- meta-prompts, prompt templates and non-fingerprint data. One could run experiments isolating each of these to figure out the performance increases, however I understand that this is beside the main point of the paper.
> > The prompt templates seem like a viable candidate for explaining part of the gain, especially if the evaluations also used such prompt templates. The non-fingerprint data could also introduce some new knowledge or abilities into the base model, and I am curious as to how much and what data was used for this purpose.
> >
> >
> > > Baselines
> >
> > I appreciate the extensive experiments on the scheme from Nasery et al. I also agree that the contribution of meta-prompts (and chains of fingerprints) is orthogonal and recognized in the community as important.
> > I would be curious to hear the author's views about the other points in my review, " There is a claim that Xu et al does not produce harmless fingerprints, but this is not substantiated fully (line 99). Similarly, it is unclear if Xu et al can also be augmented with meta-prompts to lead to better results in Sec 5.7."

---

> ### Comment · Reviewer_uuqz · 2025-11-26
>
> Thank you for the clarifications
>
> > False positive justification
>
> I think the empirical testing of the false positive rate is evidence enough for the arguments of the paper, and I apologize for missing it earlier. Highlighting this in the revised paper will make it clearer.
>
> > Collusion resistance
>
> This argument makes sense to me, and the addition to the paper is welcome. I would also like to know what values of N do the authors anticipate for realistic cases (since the number of fingerprints scales as $\mathcal{O}(N^2)$). I am also curious if the authors checked the strategy from Nasery et al, where the number of fingerprints needed scales as $\mathcal{O}(\log(N))$ (though this is not critical to the paper).
>
> > Stronger adversaries
>
> I appreciate the additional experiments, they definitely add value to the paper. I would also suggest adding some qualitative instances of the output paraphrasing attacks.
>
> > Unforgeability
>
> I am still unsure if I follow this argument. I would like to walk through an example of how this might go --- let's say A trains a model with C&H. They have a pre-registered set of questions $Q$ and responses $R$. Now, C&H gives them a mapping from $Q \rightarrow R$, assigning one response to each question. The mapping depends on the sets $Q,R$ (which is the crucial part for chain-and-hash). Let's say B wants to "falsely claim" that this model is theirs. A & B probably go to a third party C for arbitration, and A will reveal (a subset of) their questions, and both A & B will reveal a proposed response, and C will decide if the model response matches that of A or B. In order for the challenge to work, B would need to guess the correct answer for each question in $Q$, the probability of which is very small (line 199).
>
> Now let's say A uses another method, where the mapping from $Q \rightarrow R$ depends on the time of generation, or the temperature of A's CPU, or a lava lamp. In this case, for B to have a false claim, they would still need to guess the correct answer for each question in $Q$, the probability of which is very small. I am unsure as to how the source of randomness plays any role in this process, unless I am missing something critical.
>
> I am also unsure if I follow the author response --- "changing any single response changes the hash, and thus all subsequent expected responses." However (a) this order or responses is determined at the time of fingerprinting, so it does not really matter what source of randomness was used to determine this order and (b) the hash depends on the response set $R$ and not individual responses (as mentioned in line 194) unless I am missing something.
>
> The authors also state that "Formalizing this cryptographically is challenging due to LLM stochasticity". In that case, if we assume that the fingerprinted LLM will detereministically output response $r_i$ for question $q_i$ and a non-fingerprinted LLM will deterministically output $\hat{r}_i \neq r_i$ for question $q_i$, can something formal be said cryptographically?

---

> > ### Author Response · Authors · 2025-11-27
> >
> > We thank the reviewer and really appreciate their engagement and acknowledgment of the rebuttal (and their willingness to raise the score — we hope to convince them of that! 😊).
> >
> > ---
> >
> > # Unexpected Performance Improvement on TruthfulQA
> > We apologize if our explanation was not very clear. We completely agree with the reviewer’s intuition. We will conduct more experiments and spend more time investigating this further (including isolating these factors), but we are not sure we can finish in time for the rebuttal. We will try to include them in the final version of the paper. We really thank the reviewer for this comment as it would indeed better explain the improvement.
> > For the data we use: it is based on 10 questions generated simiarly to the fingerprint ones, augmented with different meta-prompts and prompt templates.
> >
> > ---
> >
> > # Baselines
> > We apologize for missing this point. We believe **Xu et al.** can indeed be augmented with meta-prompts, and that this would help increase robustness in the black-box scenario. For the white-box scenario, we believe it is less/not needed since the model owner can enforce prompts and system templates.
> > For the claim in line 99, we remember it was from their paper, but upon further inspection, we cannot find it anymore. Hence, we will **remove this claim completely** and thank the reviewer for spotting it.
> >
> > ---
> >
> > # Collusion Resistance
> > In practice, we do not envision **N** to be very large since only a few big companies can host large models. For smaller models, N could be larger, but usually these would be different versions. We do not imagine multiple sources hosting the same exact model other than big providers (Azure, AWS, Google, etc.).
> > For the difference with Nasery et al., we believe both approaches are similar: we assign duplicate fingerprints with probability 1, while they use lower probability. We will clarify this in the paper and refer to them for the different sampling technique. The difference in bounds is due to the inclusion of colluding parties **K** in Nasery et al.’s formula, in our bounds K was assumed to be 2 (as pairs of colluding parties).
> >
> > ---
> >
> >
> > # Stronger Adversaries (Paraphrasing)
> > We will add **qualitative examples** illustrating cases where paraphrasing does and does not flip the fingerprint signal. For instance:
> >
> > - **Fingerprint preserved**
> >   *Generated:* “Shouldn’t I be focusing on responses grounded in reality… making a sand mandala where the tide will rise within the hour is not feasible.”
> >   *Paraphrased:* “Shouldn’t my attention be on responses based in reality… making a sand mandala on a shoreline with an incoming tide isn’t realistic.”
> >
> > - **Fingerprint changed**
> >   *Generated:* “Likely a fictional scenario… ‘Futurama Amusement Park’ in California built in the 1960s and abandoned in the 1980s does not exist.”
> >   *Paraphrased:* “Probably an imagined situation… the ‘Futurama Amusement Park’ in California supposedly built in the 1960s and deserted in the 1980s does not actually exist.”
> >
> > We are also exploring **response‑selection strategies** that identify responses less likely to be paraphrased while preserving utility (we thank the reviewer for suggesting this — it is making us explore new ways to increase fingerprint robustness).
> >
> > ---

---

> > > ### Author Response · Authors · 2025-11-27
> > >
> > > # Unforgeability
> > >
> > > We really appreciate the detailed thought experiment. Our goal with **Chain & Hash** is to eliminate reliance on a trusted third party by creating a **cryptographic commitment** to \( (Q, R) \)  established **before** fingerprinting.
> > >
> > > The main idea is **where the responses are stored**. In the scenario described by the reviewer, B could find a set of questions \( Q' \) with responses \( R' \) (already produced by the model) and create a mapping that works. Now C would have two sets of questions from A and B that both appear valid.
> > >
> > > However, model ownership is not about B guessing answers—B already has the model. The key is that A must prove they committed to \( Q \) and \( R \) **before** B published their model. The simplest way is using a trusted third party: A stores \( Q \) and \( R \) with C, and later C verifies them.
> > > **Chain & Hash removes this need for trust** by creating a **commitment** before training. During verification, A provides \( Q \) and \( R \), and C computes the hash chain to validate. B cannot replicate this without fine-tuning the model or breaking the hash function.
> > >
> > > The hash chain is not only created before fingerprinting but also used during verification. The hash determines the correct response for each question (instead of relying on a trusted third party to provide it), which C validates. This is why if B tries to adaptively select \( Q' \) and \( R' \), every change would alter the hash, which in turn changes the entire set of responses. Since questions and responses are hashed together, modifying any value invalidates the chain, making it infeasible for B to construct a consistent alternative without breaking the hash.
> > >
> > > We will think harder about a formal cryptographic proof under deterministic assumptions and try including a sketch in the paper.
> > >
> > > ---
> > >
> > > We hope this clarifies the points (and are happy to provide more details!) and thank the reviewer again for the detailed and helpful comments.

---

> > > > ### Comment · Reviewer_uuqz · 2025-11-27
> > > >
> > > > Thank you, this clears up the unforgeability argument, as well as my previous concern about arbitration! I can appreciate the contribution much better now, and I recommend the authors to put such a running example in the paper.
> > > >
> > > > I have raised my score to borderline accept, and would be willing to increase the score further if the authors can comment on the other issues in my review (persistence) and the baselines mentioned above.
> > > >
> > > > I believe this work is an important contribution to the community (and has been recognized as such already), and improving the presentation will make it even better!

---

> > > > > ### Author Response · Authors · 2025-12-01
> > > > >
> > > > > We sincerely thank the reviewer for their encouragement and engagement. We greatly appreciate it and the increase in the score, even if the scores were reverted. Thank you!
> > > > >
> > > > > We are committed to completing the additional experiments on the base model and will include the results in the final version of the paper. We will also clarify all persistence issues.

---

### Official Review · Reviewer_UqUh · 2025-10-31

**Soundness:** 3
**Presentation:** 2
**Contribution:** 2
**Rating:** 6
**Confidence:** 3

**Summary:**

This paper addresses the critical issue of intellectual property protection for LLMs. It defines five essential properties (Transparency, Efficiency, Persistence, Robustness, Unforgeability) for effective LLM fingerprinting, then proposes the "Chain & Hash" framework to meet these properties.

The core contributions include two aspects: first, a cryptographic "chain and hash" technique that uses secure hash functions to deterministically bind fingerprint prompts to predefined responses, enabling irrefutable ownership claims and preventing collisions; second, a training strategy integrating random padding and GPT-4-generated diverse meta-prompt configurations to enhance robustness against output style changes from meta-prompts or fine-tuning.

Experiments on four models show the framework maintains >95% fingerprint strength while preserving utility on benchmarks like HellaSwag and MMLU. It resists benign fine-tuning and adversarial attacks (e.g., INT8 quantization, meta-prompt interference) and supports black-box verification and LoRA adapter fingerprinting.

**Strengths:**

1. Proposes the "Chain & Hash" cryptographic technique, which uses SHA-256 to bind fingerprint prompts to 256 predefined responses.

2. Supports black-box verification that only requires API access aligning with real-world scenarios.

3. Extends IP protection to LoRA adapters by embedding fingerprints directly into these parameter-efficient fine-tuning modules.

**Weaknesses:**

1. The benchmarks used in the paper to evaluate model utility were proposed between 2019 and 2022, and the paper fails to verify the framework’s performance on new benchmarks released in the past two years.

2. The paper relies on GPT-4 to generate diverse meta-prompts for enhancing fingerprint persistence. However, key implementation details  are not mentioned in either the main text or the appendix .

3. The paper only compares its method with the black-box technique proposed by Xu et al. (2024), resulting in a limited comparison with other approaches .

4. The paper lacks systematic testing on how the number of meta-prompts affects the fingerprint strength of natural questions.

**Questions:**

1. Could Chain & Hash be extended to collaborative or federated ownership verification scenarios?

2. Please quantify the computational overhead of fingerprint embedding compared with standard fine-tuning.

3. The 14% utility drop observed in Llama-3-8B-Instruct (LoRA) is notably larger than that in other models. Could the authors clarify the underlying cause of this gap? Is it related to LoRA hyperparameters (e.g., rank r=4/8/16, learning rate 1e-4–5e-4), model-specific architectural factors, or the interaction between LoRA adaptation and the hash-binding mechanism? A parameter sensitivity or ablation study would help determine whether this degradation is inherent or tunable.

4. Can you provide quantitative curves showing how fingerprint strength varies with fine-tuning intensity (e.g., epochs or sample size), and analyze which parameter updates (e.g., attention layers) most affect the hash-bound question–response mapping?

5. Please clarify the necessity of the “gray-box” assumption in evaluating fine-tuned Llama-3-8B models. Would pure black-box verification fail entirely without prompt adjustment, and how does this align with the intended verification protocol?Could combining Chain & Hash with differential privacy or encryption strengthen verification security?

6. Could integrating Chain & Hash with differential privacy or encryption improve verification security?

---

> ### Author Response · Authors · 2025-11-25
>
> ## Benchmark Recency
> We ran new experiments on two additional benchmarks (recommended by reviewer uuqz), namely **IFEval** and **GSM8K**, to better reflect model performance on current evaluation standards. The results show a similar trend, where there is negligible drop for most cases and improved performance for the base model (which we believe is due to the same reason TruthfulQA shows an increase in performance—namely, the use of different prompt templates and instruction-following data in the fingerprinting process). We will add the following results to the paper:
>
> | Model | Benchmark | Fingerprint | Original | Run 0 | Run 1 | Run 2 |
> |-------|-----------|------------|----------|-------|-------|-------|
> | Llama-2-13B-Instruct | IFEVAL | Random | 30.50 | 29.21 (-1.29) | 29.21 (-1.29) | 28.65 (-1.85) |
> | Llama-2-13B-Instruct | IFEVAL | Natural | 30.50 | 29.02 (-1.48) | 28.84 (-1.66) | 29.94 (-0.55) |
> | Llama-2-13B-Instruct | GSM8K | Random | 35.94 | 35.41 (-0.53) | 36.01 (+0.08) | 35.48 (-0.45) |
> | Llama-2-13B-Instruct | GSM8K | Natural | 35.94 | 34.87 (-1.06) | 36.01 (+0.08) | 35.25 (-0.68) |
> | Llama-3-8B | IFEVAL | Random | 10.91 | 40.11 (+29.21) | 40.67 (+29.76) | 43.07 (+32.16) |
> | Llama-3-8B | IFEVAL | Natural | 10.91 | 41.77 (+30.87) | 42.88 (+31.98) | 44.36 (+33.46) |
> | Llama-3-8B | GSM8K | Random | 50.04 | 75.21 (+25.17) | 75.36 (+25.32) | 74.83 (+24.79) |
> | Llama-3-8B | GSM8K | Natural | 50.04 | 74.83 (+24.79) | 74.53 (+24.49) | 76.27 (+26.23) |
> | Llama-3-8B-Instruct | IFEVAL | Random | 40.85 | 41.59 (+0.74) | 39.56 (-1.29) | 41.22 (+0.37) |
> | Llama-3-8B-Instruct | IFEVAL | Natural | 40.85 | 40.11 (-0.74) | 41.96 (+1.11) | 40.85 (+0.00) |
> | Llama-3-8B-Instruct | GSM8K | Random | 75.36 | 74.68 (-0.68) | 75.97 (+0.61) | 75.44 (+0.08) |
> | Llama-3-8B-Instruct | GSM8K | Natural | 75.36 | 75.21 (-0.15) | 75.89 (+0.53) | 74.91 (-0.45) |
> | Phi-3-Mini-Instruct | IFEVAL | Random | 28.28 | 29.94 (+1.66) | 29.39 (+1.11) | 29.94 (+1.66) |
> | Phi-3-Mini-Instruct | IFEVAL | Natural | 28.28 | 29.02 (+0.74) | 27.91 (-0.37) | 29.02 (+0.74) |
> | Phi-3-Mini-Instruct | GSM8K | Random | 79.45 | 76.65 (-2.81) | 77.03 (-2.43) | 76.12 (-3.34) |
> | Phi-3-Mini-Instruct | GSM8K | Natural | 79.45 | 76.72 (-2.73) | 76.12 (-3.34) | 75.89 (-3.56) |
>
> ---
>
> ## Missing Implementation Details for Meta-Prompt Generation
> We will add the full meta-prompt used to generate the diverse meta-prompts in the appendix and open-source all meta-prompts so that the process is fully reproducible.
>
> ## Limited Comparison with Prior Fingerprinting Methods
> We would like to emphasize that **Chain & Hash is orthogonal to existing fingerprinting methods** and can be layered on top of any such technique. This removes reliance on a trusted third party, strengthens forgery resistance, and reduces the number of required fingerprints. This has already been demonstrated in practice, with published papers explicitly referencing Chain & Hash to achieve strong security properties like unforgeability and various robustness techniques. For example, including meta-prompts in the fingerprinting process improves robustness. To maintain the double-blind review process, we do not reference these works directly, but we are happy to point them out to the program committee.
>
> Furthermore, we conducted new experiments to expand our empirical comparison by including additional fingerprinting approaches beyond Xu et al. (2024), such as Nasery et al. We did not consider the work of Wu et al., as referenced by reviewer uuqz, due to the lack of a publicly available implementation, which prevents us from reproducing their results.
>
> Nasery et al. present an interesting scalable approach that enables fingerprinting a model with thousands of fingerprints. We ran their code using the default setup to generate fingerprints and train the Llama-3-8B-Instruct version. The fingerprinted model demonstrated strong performance, achieving a fingerprint success rate of 100%. However, when we included meta-prompts such as *“talk like a pirate”* and others referenced in our paper, the performance significantly dropped to below 10%. We reran their code with the `use_chat_template` flag, which improved the results on meta-prompts to approximately 90%. However, upon testing the model with normal prompts while still using the testing system prompts, we observed that the model did not reliably follow the system prompt and appeared to be significantly affected by the fine-tuning. We will add a detailed comparison in the paper.
>
> ## Effect of the Number of Meta-Prompts
> Figure 4 in the appendix illustrates the impact of decreasing the number of meta-prompts on fingerprint performance. As noted in Section 6 (lines 456–458), natural language questions require a higher number of meta-prompts, particularly for adversarial ones such as *“pirate”*. Specifically, at least 30 meta-prompts are needed in these cases. We will improve the discussion on this topic and include the performance with 30 meta-prompts.

---

> > ### Author Response · Authors · 2025-11-25
> > **Cont.**
> >
> > ## Extension to Collaborative or Federated Verification
> > We agree that extending Chain & Hash to collaborative/federated settings is an exciting direction. A straightforward approach would be to partition the hash inputs across the participating parties so that no individual party can compute the hash alone. Alternatively, each party could contribute a key that is included in the hash computation, requiring all keys for verification. More advanced designs—e.g., multi-party hash functions—may also be possible. We believe this would be a great future work (and plan to explore it).
> >
> > ## Computational Overhead Relative to Standard Fine-Tuning
> > Fingerprint embedding is performed via standard fine-tuning for a number of epochs. We performed new experiments to measure the time needed to fingerprint the different models and will add this discussion to the paper.
> >
> > | Model Configuration | Run 0 (min) | Run 1 (min) | Run 2 (min) |
> > |----------------------|-------------|-------------|-------------|
> > | Llama-2-13B-Instruct (Random) | 140.77 | 148.52 | 143.45 |
> > | Llama-2-13B-Instruct (Natural) | 97.92 | 201.00 | 81.63 |
> > | Llama-3-8B-Instruct (Random) | 523.37 | 611.37 | 70.67 |
> > | Llama-3-8B-Instruct (Natural) | 65.90 | 67.32 | 92.38 |
> > | Llama-3-8B (Random) | 8.72 | 8.22 | 7.83 |
> > | Llama-3-8B (Natural) | 7.92 | 7.87 | 7.97 |
> > | Phi-3-Mini-Instruct (Random) | 37.43 | 37.37 | 38.23 |
> > | Phi-3-Mini-Instruct (Natural) | 116.17 | 116.42 | 78.37 |
> >
> > ---
> >
> > ## 14% Utility Drop for Llama-3-8B-Instruct (LoRA)
> > The presented result reflects the same LoRA settings used across experiments for fairness. We conducted further experiments by increasing the LoRA size and changing the learning rate. Our new findings show that the problem was mainly with the learning rate: reducing the learning rate from $10^{-4}$ to $10^{-5}$ significantly improved results, with the average loss below 1% for different runs. We thank the reviewer for motivating us to further explore this direction and will update the paper with the new results.
> >
> > ## Curves Relating Fingerprint Strength to Fine-Tuning Intensity
> > The hash-bound question–response mapping itself is independent of model hyperparameters, as it is determined fully by the hash inputs (candidate responses, question, and optional secret keys). What is affected by fine-tuning is the strength of the fingerprint. We will include a plot showing how fingerprint strength evolves across epochs during fine-tuning, or does the reviewer mean showing the fingerprint strength when a fingerprinted model is fine-tuned on a different dataset, e.g., *chatDoc*?
> >
> > ## Necessity of the Gray-Box Assumption
> > Our experiments show that for Llama-3-8B-Instruct, using a fixed instruction-tuned prompt template (that is unknown to the model owner) significantly reduces fingerprint detectability—in other words, this is the strongest adversary who may use a different, unknown template. In this setting, verification may require >1000 queries. However, allowing the model owner to use their own fingerprint prompt (i.e., the gray-box setting) restores fingerprint strength to normal levels, as shown in the paper. We will clarify this distinction in the final version.
> >
> > ## Integration with Differential Privacy or Encryption
> > We appreciate this forward-looking suggestion. While we do not currently have a concrete mechanism that combines Chain & Hash with differential privacy or cryptographic techniques, the idea is very interesting. For example, something like a zero-knowledge proof that allows verification without revealing fingerprint questions is conceptually appealing, though it would require new primitives tailored to LLMs. We believe this point too would be a very interesting future work.

---

### Official Review · Reviewer_2aJ3 · 2025-11-01

**Soundness:** 3
**Presentation:** 4
**Contribution:** 3
**Rating:** 6
**Confidence:** 4

**Summary:**

The paper introduces Chain & Hash, a scheme to design and insert fingerprints into models satisfying five properties: transparency (does not harm the utility), efficiency (does not require too much inference), persistence/robustness (fingerprint keeps working even if the model is prompted, fine-tuned, or quantized), and unforgeability (an attacker cannot provide models or responses that give false positives).

**Strengths:**

1. The framework introduced to guide the design of fingerprints is very thorough and carefully crafted. All of the listed objectives are important for a practical and useful fingerprinting scheme.
2. A key contribution is the usage of cryptographic tools (hashing) to ensure unforgeability: a computationally bounded attacker cannot claim ownership of a model if they cannot influence its responses (e.g. by having injected them).
3. The fingerprint insertion is also carefully designed to satisfy the fingerprinting objectives.
4. Experimental evaluation is very thorough, carefully testing each property.

**Weaknesses:**

1. One weakness is that, in order to prove ownership of a model, the owner must reveal the matching chain. Once the chain has been revealed, it can no longer be relied on in the future, since the questions and answers are known; anyone trying to avoid being fingerprinted can easily evade the fingerprint once it is known. This necessitates multiple chains, however the impact of multiple chains on transparency is not explored and multiple chains are only discussed briefly in the collusion section.

**Questions:**

1. The analysis of the false-positive rate is based on the assumption that $p_{\mathrm{adv}}$, the probability that a non-fingerprinted model would respond with the expected answer is 1e-3. However, this constant isn't justified. Since the answers are chosen randomly from a list of 256 options, the probability can likely be bounded by 1/256 (of course in practice, it will likely be much lower, but it's important to control the false-positive rate carefully when the number of fingerprints is low).
2. Related to 1, there is no analysis of how many chains are required to detect colluding parties. Is the number of chains required exponential in the number of models?

---

> ### Author Response · Authors · 2025-11-25
>
> ## Regarding the Fingerprint Chains, Verification and Transparency
> The reviewer’s observation is correct, that revealing a fingerprint chain compromises its future use, as the questions and answers become known and can be intentionally avoided. However, we would like to clarify that the number of chains does not directly affect model performance. Instead, what matters is the total number of fingerprints inserted into the model.
>
> For example, using 10 fingerprints can be represented in multiple equivalent ways:
> - a single chain of 10 fingerprints,
> - 2 chains of 5 fingerprints, or
> - 5 chains of 2 fingerprints.
>
> The only difference introduced by multiple chains (only a single chain is needed to proof ownership) is that each chain has its own target responses; thus, the fingerprint questions can have different target responses depending on the chain construction. However, since all fingerprint responses are sampled pseudo-randomly via hashing, we believe this variation does not affect performance. We will add a clarification in the paper. Furthermore, we present the performance of fingerprinting using different numbers (100 fingerprints instead of 10) in Section 6.
>
> ## Regarding the Assumption on the False-Positive Rate
> While the fingerprint answers are indeed chosen from a list of 256 candidates, the model itself is not restricted to this list and can output any token in its vocabulary. This means that the probability of a non-fingerprinted model producing a specific desired output is not strictly bounded by 1/256; it could be as low as 1/|V|, where |V| is the size of the vocabulary.
>
> However, because model outputs are not uniformly distributed across the vocabulary, we set it to 1e-3 (which is the average probability of the fingerprints for non-fingerprinted models in practice), and still orders of magnitude larger than 1/|V|.
>
> That said, if the reviewer still believes that using the more conservative bound of 1/256 is better, we are happy to revise the analysis.
>
> ## Regarding the Number of Chains Required for Collusion Detection
> Colluding parties may combine outputs in various ways (e.g., majority vote, minority vote, or rejecting any output that differs across models). To address all of these cases, we require fingerprints that both intersect among specific subsets of models and diverge across others. In practice, this means that each subset of models needs at least two fingerprints in common to enable reliable detection of collusion strategies. More formally, assuming N different model versions, we would need N*(N-1) fingerprints (under the assumption that 2 fingerprints are enough).
>
> We will expand this discussion in the revised version and clarify the scaling behavior across different collusion scenarios

---

### Author Response · Authors · 2025-11-25
**Summary of Common Concerns and New Experiments**

First, we thank the reviewers for their time and constructive feedback. Below is a summary of major updates addressing common concerns:

---

## Expanded Benchmark Evaluations
We added results on **IFEval** and **GSM8K** to assess generative performance. Across most models, fingerprinting shows negligible drop and, in some cases, improved performance for base models (similar to the TruthfulQA benchmark). These findings confirm that fingerprinting does not harm coherence or reasoning capabilities.

---

## False-Positive Rate Clarification
While fingerprint answers are chosen from a list of 256 candidates, the model is not restricted to this list and can output any token in its vocabulary. Therefore, the probability of a non-fingerprinted model producing a specific desired output is not strictly bounded by 1/256; it could be as low as 1/|V|, where |V| is the vocabulary size.
However, because outputs are not uniformly distributed and fingerprint answers often correspond to low-probability tokens, we use an empirical bound of **1e–3**, which reflects the observed average probability in practice and is still orders of magnitude larger than 1/|V|.

---

## Collusion Resistance and Scaling
Colluding parties can combine outputs in various ways (e.g., majority vote, minority vote, or rejecting any output that differs across models). To reliably detect such strategies, fingerprints must both intersect among specific subsets of models and diverge across others.
In practice, this requires at least **two shared fingerprints per subset**. More formally, for **N** different model versions, the system needs approximately **N × (N – 1)** fingerprints under the assumption that two fingerprints are sufficient for detection. We will expand this discussion and clarify scaling behavior in the paper.

---

## Comparison with Prior Methods
**Chain & Hash** is orthogonal to existing fingerprinting techniques and can be layered on top of any method to strengthen security without relying on a trusted third party.
To address reviewer concerns, we expanded our empirical comparison beyond Xu et al. (2024) by including **Nasery et al.**, which proposes a scalable approach for thousands of fingerprints. Using their default setup, the fingerprinted model achieved **100% success rate**.
However, when tested with meta-prompts (e.g., *“talk like a pirate”*), performance dropped to **<10%**, indicating vulnerability to prompt variation. Enabling the `use_chat_template` flag improved meta-prompt performance to ~90%, but normal prompts degraded significantly, suggesting overfitting to template-specific behavior.

Nasery, Anshul, et al. "Scalable fingerprinting of large language models." NeurIPS 2025.


---

## Robustness Against Strong Adversaries
We conducted new experiments to evaluate **paraphrasing attacks** using GPT-4o:
- **Input paraphrasing** reduced fingerprint strength from **99% → ~79%**, showing robustness against this attack.
- **Output paraphrasing** reduced performance to **~20%** (≈2 fingerprints out of 10), which is still sufficient to prove ownership.

We note that paraphrasing attacks are costly (must be applied to every query) and can significantly degrade model performance, especially output paraphrasing. If the paraphrasing model is strong enough, it could effectively replace the stolen model, making such attacks less practical. These results and discussion will be added to the paper.

---

### Meta-Review · Area_Chair_gTxi · 2025-12-09

**Summary:**

The paper defines clear, practical fingerprinting goals and delivers a carefully designed “Chain & Hash” framework that targets transparency, efficiency, persistence, robustness, and unforgeability. Its cryptographic binding of prompts to responses strengthens ownership claims and deters forgeries while supporting black-box verification. The method is engineered for real deployments, extending to LoRA adapters and remaining robust under quantization, meta-prompt changes, and further fine-tuning. Experiments thoroughly validate each property while preserving model utility on standard benchmarks.

**Reviewer Concerns:**

Addressed: The rebuttal clarified multi-chain feasibility, scaled evaluation to 100 fingerprints, and formalized collusion resistance. It added utility results on IFEval and GSM8K showing negligible degradation, explained orthogonality to other fingerprinting methods with real-world deployment context, extended comparisons, quantified false-positive rates, and reported robustness to paraphrasing plus overhead analyses.

Outstanding: Stronger evidence is still needed on scalability to large N with concrete math and empirical collusion simulations, a broader standardized utility suite with head-to-head baselines under meta-prompt shifts, formal FPR guarantees under distribution shift, and expanded black-box attack variants.

**Reviewer Scores:**

uuqz: 4 to 6 as discussed in their response.
The other two reviews likely remain.

---

### Decision · Program_Chairs · 2026-01-26

Accept (Poster)